# Integrative review in PhD admissions: A case study of efficiently minimizing bias while maximizing the student narrative

Minerva A. Orellana[1,2☯], Danielle J. Beetler[3,4☯], Carmen J. Silvano[3], Ryan Wuertz[3], Jennifer L. Weisbrod[3], Lewis R. Roberts[3,5], Anthony J. Windebank[3,6], Felicity T. Enders[2,3], Marina R. Walther-Antonio[3,7,8,9]*

1 Department of Obstetrics and Gynecology, University of Washington, Seattle, Washington, United States of America, 2 Department of Quantitative Health Sciences, Mayo Clinic, Rochester, Minnesota, United States of America, 3 Center for Clinical and Translational Science, Mayo Clinic, Rochester, Minnesota, United States of America, 4 Department of Cardiovascular Medicine, Mayo Clinic, Jacksonville, Florida, United States of America, 5 Division of Gastroenterology and Hepatology, Mayo Clinic College of Medicine and Science, Rochester, Minnesota, United States of America, 6 Department of Neurology, Mayo Clinic, Rochester, Minnesota, United States of America, 7 Department of Obstetrics and Gynecology, Mayo Clinic, Rochester, Minnesota, United States of America, 8 Department of Surgery, Mayo Clinic, Rochester, Minnesota, United States of America, 9 Microbiomics Program, Center for Individualized Medicine, Mayo Clinic, Rochester, Minnesota, United States of America

☯ These authors contributed equally to this work.
* WaltherAntonio.Marina@mayo.edu

## Abstract

Developing scientific and medical innovations continue to be limited by lack of diverse representation among leaders and learners. One key gateway for these goals is graduate school admissions, but comprehensive consideration of all components of applications, which is needed to reduce systemic bias in admissions, is resource intensive. This case study details the conceptualization of an integrative application review process to challenge and improve classic application review frameworks which gatekeep admissions opportunities from under-represented (UR) applicants. PhD applicant cohorts to a longstanding Clinical and Translational Sciences PhD TL1 program were assessed using one of three review processes: traditional, algorithmic, or a novel integrative review process. Admissions results from each review process were pooled across matriculation years to attain a testable sample size. Effects modification models were used to assess odds of reaching each admissions phase, adjusting for UR status and review process. Results showed that classic admissions review processes were prone to bias towards admission of specific students while integrative application review did not demonstrate this trend. The Mayo Clinic Graduate School of Biomedical Sciences Clinical and Translational Sciences training program has steadily recruited and trained successful and diverse trainee cohorts over the last decade from many underrepresented backgrounds. The final adoption of an integrative application review process allows streamlined graduate school admissions

**Data availability statement:** The Mayo Clinic IRB board deemed this study (IRB 24-003992) exempt because it was a secondary data analysis using de-identified information and did not require participants' consent. However, this data cannot be made publicly available because it is protected by the Family Educational Rights and Privacy Act (FERPA), a federal law protecting the privacy of student education records. Cohort data consists of small sample sizes that can potentially allow the identification of participants. Data requests can be sent to the Center for Clinical and Translational Science Education Program at CCaTSEDUProg@mayo.edu.

**Funding:** This publication was supported by CTSA Grant Number TL1 TR002380 from the National Center for Advancing Translational Science (NCATS). Its contents are solely the reponsibility of the authors and do not necessarily represent the official views of the NIH. The funders had no role in study design, data collection and analysis, decision to publish, or preparation of the manuscript.

**Competing interests:** The authors have declared that no competing interests exist.

of diverse student cohorts, prioritizing self-driven narratives and minimizing subjective biases where possible to allow fair assessment of learners.

## Introduction

### Impact of admissions in higher education in the United States

As of 2020, 61% of biomedical science graduate students self-reported as 'Non-Hispanic White' [1]. Ethno-racially diverse individuals are underrepresented as learners despite decades of programs and initiatives to address this disparity, perhaps in part due to a lack of consensus on ideal strategies [2]. Hispanic/Latinos represented 19% of the US population in 2022 but only 8% of those receiving doctorates and 12% of graduate students. Likewise, Black people accounted for 13.6% of the population, yet only 6% of doctoral graduates and 6% of graduate students [3,4]. Moreover, race and ethnicity metrics represent only the most accessible diversity data, with many facets of identity varying across learners, including familial education background, gender identity, and socioeconomic status, etc. Minority status in any one or combination of these identities can result in educational opportunity disparities, especially in higher education cohorts.

Learners with access to diverse peer cohorts and faculty mentors have greater opportunities for scientific impact, facilitating cross-cultural influence [5]. Not only are learners affected by diversity or lack thereof in the learning environment, but educational institutions at the forefront of modern medicine are similarly affected by these demographics. Scientific and medical innovations continue to be limited by a lack of diverse representation among leaders and learners. Clinical trials disproportionately under-represent people of color despite repeated calls to action and changes in recruitment and funding standards by directing bodies like the National Institutes of Health (NIH). Increasing representation of diverse clinical research personnel can help mitigate this [6]. Physicians from underrepresented groups often work with their underserved or under-represented communities, expanding service and access to those communities [7]. It is essential to diversify both the medical professionals that administer clinical trials/care *and* the scientific personnel that design this research to improve care standards, practices, and policies.

It is important to attract, develop, and support the best scientists from all groups and at all career stages to increase the implementation of new ideas, methodologies, and approaches [8,9]. The National Center for Advancing Translational Science (NCATS) has recognized the value of a diverse workforce to advance clinical and translational research [10], which resulted in Clinical and Translational Science Award (CTSA) programs and Clinical and Translational Science Institutes (CTSIs) prioritizing diversification by emphasizing recruitment of diverse scientists. One key gateway for these goals is graduate school admissions.

### Components of graduate school application

Graduate school applications consist of multiple components: grade point average (GPA), Graduate school Readiness Exam (GRE), personal statement (PS), letters

of recommendation (LOR), and *curriculum vitae* (CV). Certain controversies exist about inclusion of each of these components in application review [11]. We provide a brief overview of each element with our stance on its relative proclivity to bias as background for the framework developed by our admissions team.

The *GRE* is an American standardized test designed to predict graduate school readiness. As an almost universal element to assess American graduate school readiness, the GRE has a long history of contests to its fairness, validity, and bias [11]. Along with systemic inequalities in scores across groups and in the availability of preparatory resources [12–14], it is not clear if high GRE scores demonstrate high certainty in successful graduate careers [14–20].

*GPA* mirrors the quantitative quality of the GRE and is associated with similar concerns. Moreover, some institutions rely on capped systems that weigh percentage grades differently to limit top-heavy distributions, which can disproportionately affect GPAs from certain schools. Importantly, GPA has a high capacity for bias for socioeconomic status, which determines both access to high schools and GPA in college [15,21]. Undergraduate GPA is correlated with graduate GPA [16], but again this does not necessarily translate to successful graduate careers [11,15,16,20].

The *PS* is the first of several qualitative elements that are almost universally required in graduate school applications. Admissions personnel can use personal statements to evaluate writing ability, program and institutional fit, and research match [11,22], although this is subject to the evaluator's impression. Some work has measured the predictivity of personal statements on graduate student potential and/or success, revealing limited validity and standardization [11,12,23]. Despite these limitations, the PS is the only opportunity for individual narratives to be reflected in an applicant's own words.

The *CV* details the applicant's credentials through a professional career summary. For graduate schools, the CV serves as the main demonstration of prior research experience, one of the highest weighted metrics by PhD program admissions teams. However, CVs list institutional names and sometimes principal investigators of projects, potentially leading to biases regarding institutional or individual researcher prestige. Moreover, the CV may demonstrate access to opportunities rather than competency and thus could have much stronger negative impacts on already underrepresented applicants [24–26]. Studies predicting graduate student success show that CVs should be utilized and interpreted with caution [15,23,27]. We encourage application reviewers to consider the CV under a careful lens, considering what the applicant has gained from each project or experience rather than weighing prestige or even amount or duration of opportunities.

*LOR* serve as an important outside evaluation of applicants. Evaluations of LOR show a correlation to PS, perhaps demonstrating the similarity of these open-ended application elements [11]. However, there are issues with LOR that make them especially prone to bias. Both the letter writer and the evaluee's gender, race, nationality, and culture may all affect the content and style of messaging, which also biases evaluators and admissions committees [28–38]. The applicant is also vulnerable to the absolute weight that can be placed on a single mentor's opinion, furthering imbalances of power. Issues of access to letter writers may be an additional area of inequity, but since a wide range of people can assess an applicant, this is less of a concern.

## Current admissions practices

Currently, there are three major admissions framework styles: traditional, algorithmic, and holistic. These are not necessarily independent frameworks, as they can merge at specific points of the admissions process. Traditional admissions are extremely individualized to each institution and consider any of the widely available application elements: GPA, GRE, PS, LOR, and CV. We agree with prior cautions that this style is most prone to biases that may limit desired demographic changes and equitable access for all applicants [11]. Algorithm-style admissions incorporate rankings of application elements into pre-determined formulas that allow customized weights for each application component based on program determination. While these are lauded as more reproducible and reduce chances of final selection changes due to unequal manipulation [11,16], they are still influenced by application elements whose rankings are prone to bias, thus biasing at least a part of the composite score. Holistic admissions attempt to consider an entire applicant by looking at all or most of the application elements and deciding on an applicant's overall potential in a non-metrics-based fashion. While this consideration of individual applicants' narratives is more comprehensive, these methods are labeled as too time and

resource intensive to truly integrate into mainstream admissions [36]. Often, this holistic approach is used secondary to an algorithmic or metrics-based screen, which can eliminate many qualified applicants before they can be considered holistically. Despite the importance of the application process, applicant characteristics presumed to predict success in the biomedical sciences are based largely on untested assumptions [15].

An important concept that has been heavily referenced here and in other works of similar design is that of a 'successful' graduate student. The metrics most often considered are time-to-degree (TTD) and publication number or author position, but others include degree attainment, citation counts, and presentations during the graduate school career, which are all indicative of professional graduate student productivity [22]. However, these 'success' metrics are not necessarily guarantors of later career success, satisfaction in the program, or personal happiness, which are all important factors to students, alongside research productivity [39]. Additionally, these metrics can be influenced by factors outside of student control: principal investigator support, collaborator access, global pandemics, socioeconomic factors, and serendipity, to name but a few [15,40]. Studies have shown that noncognitive factors (resilience, work ethic, maturity, organization skills, etc.) are also important [22,40]. In recruiting the 'best' students during each admissions cycle, individual graduate schools and/or programs must consider and define what a successful graduate of their program is. In our program, we advocate heavily for collaboration and team science.

### Framework for an integrative application review

While no application element provides a perfect assessment of graduate student success, several contribute to systemic exclusion for certain groups [41]. Our team's goal is to implement and share an admissions framework that streamlines the application review process while preventing the loss of diversity in admitted graduate student cohorts. This case study will detail the evolution of our review process, culminating in our final adoption of an integrative application review process, which incorporates review of the applicant's narrative and experiences without subjective application components.

## Methods

### Integrative application review process

The Mayo Clinic Graduate School of Biomedical Sciences (MCGSBS) Clinical and Translational Sciences (CTS) Doctor of Philosophy (PhD) program has continually adapted its admissions committee and process. Initially, we incorporated a traditional review process (2008–2018) and then transitioned from an algorithmic review process (2019–2021) to our current integrative approach (2022–2024) (**Fig 1**). This integrative approach involves reviewing only the PS to focus solely on the applicant narrative instead of considering all application components in the initial review. One committee member maintained access to the complete application (unblinded with all components) to verify that the applicants met initial MCGSBS requirements (minimum 3.0 GPA, Bachelor's in Biological Science, three letters of recommendation, CV, and unofficial transcripts). The admissions committee, composed of faculty, alumni, current students, and staff (n = 14) completed mandatory implicit bias training prior to reviewing applications and interviewing applicants. PS were removed from the application and de-identified before storage in a centralized location. Reviewers were assigned approximately 25 applications (de-identified PS only) for review using a vetted rubric, with each applicant having seven reviewers (1–2 from each program role included in the admissions committee). The rubric included open-ended prompts for the reviewer to consider while reading the PS, such as institution and program match and evidence of drive/passion. Each reviewer ranked applications and answered a question of whether they would recommend the applicant for interview: definitely, yes, maybe, and no. The program coordinator presented aggregate results at a committee meeting where reviewer consensus was reached. Applicants that received all 'definitely' and/or 'yes' votes on the recommend to interview question were added to the interview list and discussions followed regarding applicants with any 'maybe' or 'no' votes. These discussions were facilitated by reviewer feedback from the rubrics. At the time of the interview, interviewers received all de-identified application components. For

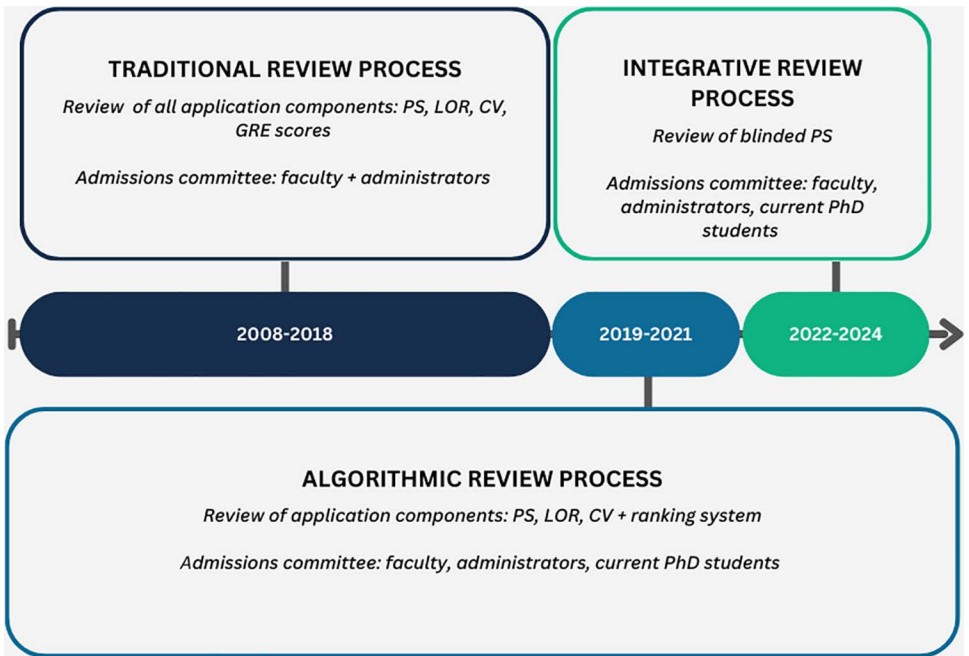

**Fig 1. History of the MCGSBS CTS PhD committee and review process.**

transparency, interviewees (total n = 56) were made aware that they were selected based on the review of PS alone. Following interviews, committee members discussed the top interviewees who received admission offers (total n = 20).

## Key terminology

We define students from under-represented groups (UR) using the NIH definition [42]. This includes students from under-represented racial and ethnic groups, students with disabilities, and students from disadvantaged backgrounds, including socioeconomic and educational.

## Data analysis

Descriptive methods were used to describe the composition of the MCGSBS CTS students. Admissions data was available starting in 2008. Prior to 2014, only ethno-racial identity was collected as a UR measurement. Data was pooled by the phases of the admissions process to increase the sample size. Each phase of the admissions process (interviewed, offered, accepted) was evaluated using logistic regression. We include effects modification models with UR status and application review process (traditional, algorithmic, and integrative) and their interaction to assess the impact of the evolving review process on the chances of admission by UR status for each admissions phase.

The Mayo Clinic IRB board deemed this study (IRB 24–003992) exempt because it was a secondary data analysis using de-identified information and did not require participants' consent. However, this data cannot be made publicly available because it is protected by the Family Educational Rights and Privacy Act (FERPA).

## Results

Of the many under-represented identities we were interested in, national data reporting is most robust for racial and ethnic identities. From 2011 to 2020, the percentage of ethno-racially underrepresented students (Black non-Hispanic,

Hispanic, and/or American Indian/Alaska Native) climbed from 13 to 18 percent, while the percentage of Non-Hispanic White graduate students decreased closer to the national population percentage of 59.3% over the same time period [1]. CTSA TL1 programs have consistently demonstrated enrollment of more diverse cohorts, staying higher or at the highest end of these national ranges over the same decade [38]. While the yearly enrollment for any individual PhD program is small, data show that the MCGSBS CTS track, the earliest CTSA TL1 program, has steadily contributed to graduate student ethno-racial diversity from inception to present (**Table 1**), including over this well-studied decade (2011–2020).

To assess whether our admissions process had certain steps that may have acted as barriers to admission for students of under-represented identities, we analyzed the odds of individuals reaching each phase of the admissions process (i.e., getting an interview, getting an admissions offer, accepting an admissions offer) using effect modification models that analyzed the effects of variables including UR status and review process used (**Table 2**). To accommodate low sample size in individual years and to best model review processes as a variable, we pooled data for years that used the same review process. Although the earliest admissions process used (traditional review, n = 166) is outdated (admissions years 2008–2018), we include its analysis against its successor (algorithmic review process, n = 54) here to analyze our dynamic review process over time.

The effect modification models for these historical cohorts showed that there was a preference for UR students at all admissions stages. The odds of receiving an interview, receiving an offer post interview, and notably, accepting that offer was twice as high for under-represented students, a significant result ($p < 0.05$). However, these models also showed that this was not review process-specific, as this was not driven by an interaction between review process and UR status. In sum, while traditional review and algorithmic review have differences in concept and in practice, UR applicants remained more likely to gain admission.

To examine the effects of the integrative review (n = 102) process on admissions phases, compared to its predecessor, the algorithmic review process (n = 54), we used the same design to create effect modification models (**Table 3**). When comparing the integrative and algorithmic processes in a model that included effect modification with UR students, the differential effect for UR compared to non-UR students was no longer statistically significant, unlike when we compared the algorithmic review process to a traditional review process. This was true at all admissions stages. In addition, the interaction term between integrative review and UR status also lacked statistical significance, showing there was no added impact of the integrative review for UR students. We observed that students were more likely to receive interviews with the

**Table 1. Under-represented backgrounds in MCGSBS CTS graduate student enrollment by year.**

| Matriculation Year n (%) | Total (n) | Under-Represented | Matriculation Year n (%) | Total (n) | Under-Represented |
|---|---|---|---|---|---|
| 2008* | 2 | 2 (100%) | 2017 | 4 | 1 (25%) |
| 2009* | 3 | 2 (67%) | 2018 | 4 | 2 (50%) |
| 2010* | 6 | 0 (0%) | 2019 | 4 | 3 (75%) |
| 2011* | 5 | 1 (20%) | 2020^ | 6 | 4 (67%) |
| 2012* | 4 | 2 (50%) | 2021^ | 3 | 3 (100%) |
| 2013* | 4 | 2 (50%) | 2022^ | 7 | 4 (57%) |
| 2014 | 4 | 3 (75%) | 2023^ | 6 | 6 (100%) |
| 2015 | 3 | 0 (0%) | 2024^ | 11 | 5 (45%) |
| 2016 | 4 | 2 (50%) | | | |

*Prior to 2014, data was not collected for disadvantaged background (economic or educational) and disability status.

^After 2020, international applicants were considered due to additions to funding, but they are not represented here as they were ineligible for NIH funding and are not categorized in Under-Represented backgrounds.

**Table 2. Traditional versus algorithmic process.**

| Interviewed | Odds Ratio | p-value |
|---|---|---|
| *Baseline* | 1.05 | 0.78 |
| *UR* | 2.93 | 0.005 |
| *Algorithmic Process* | 2.17 | 0.11 |
| *Interaction* | 0.51 | 0.36 |

| Offered | Odds Ratio | p-value |
|---|---|---|
| *Baseline* | 0.3 | 0 |
| *UR* | 2.11 | 0.04 |
| *Algorithmic Process* | 0.5 | 0.29 |
| *Interaction* | 1.51 | 0.62 |

| Accepted | Odds Ratio | p-value |
|---|---|---|
| *Baseline* | 0.27 | 0 |
| *UR* | 2.14 | 0.04 |
| *Algorithmic Process* | 0.55 | 0.37 |
| *Interaction* | 1.49 | 0.63 |

**Table 3. Algorithmic versus integrative process.**

| Interviewed | Odds Ratio | p-value |
|---|---|---|
| *Baseline* | 2.29 | 0.07 |
| *UR* | 1.5 | 0.52 |
| *Integrative Process* | 0.34 | 0.05 |
| *Interaction* | 1.01 | 0.99 |

| Offered | Odds Ratio | p-value |
|---|---|---|
| *Baseline* | 0.15 | 0.002 |
| *UR* | 3.17 | 0.11 |
| *Integrative Process* | 2 | 0.34 |
| *Interaction* | 0.36 | 0.24 |

| Accepted | Odds Ratio | p-value |
|---|---|---|
| *Baseline* | 0.15 | 0.002 |
| *UR* | 3.17 | 0.113 |
| *Integrative Process* | 2 | 0.34 |
| *Interaction* | 0.33 | 0.2 |

integrative process. However, this could be due to the fact that the number of interview slots increased at the same time that the integrative review process was implemented.

Our summative observation from the analysis of these more current cohorts is that there were changes in the review process that eliminated the preference for UR status, but this was not specific to the integrative review process. We purport that this exemplifies the limitations of segmenting an evolving process and that perhaps subconscious or other changes may have contributed to these results.

## Discussion

### Diversity in MCGSBS CTS

Nationally, graduate student diversity is increasing. While ethno-racial diversity is a long-standing metric with historical infrastructure, other identities are underrepresented in the scientific community and offer similar benefits to the field when they are represented by its learners and leaders. As the NIH definition of an under-represented or diverse student has evolved, so has the MCGSBS CTS program's inclusion of these factors in its metrics. Our definition of underrepresented students now includes ethnic and/or racial minority group identities, first generations in college, inter-abled people, and disadvantaged socioeconomic and/or educational backgrounds, a list sure to grow.

In this case study of longitudinal analyses during changes to our admissions process, no step was observed to create a multi-year barrier to admission for UR applicants, contrary to what has been historically observed between applying and interviewed students nationally [14]. The MCGSBS CTS program consistently had a high enrollment of diverse learners regardless of the admissions review process (**Table 1**). This is encouraging, as these identities are not explicitly considered to progress through our admissions process. Additionally, there are no stark differences in trends between admissions steps, most importantly offers versus acceptances of offers. This highlights that the MCGSBS CTS program can effectively recruit diverse students to apply and retain them in attendance, suggesting an affirming environment for UR students and contributing to the NCATS strategic plan of cultivating a multifaceted, highly skilled, and inclusive translational science workforce [10].

While traditional review and algorithmic review have differences in concept and in practice, UR applicants remained more likely to gain admission. This statistically significant difference in favor of UR applicants may stem from many potential sources. A highly likely possibility is that UR applicants demonstrated better preparation than non-UR applicants. This is supported by literature suggesting that UR individuals must achieve more to be considered or even to feel that they may be considered at the same level as non-UR individuals [43,44]. Another possibility is that reviewers were biased in favor of UR applicants. Due to the nature of the data, we are unable to determine which is most probable.

Our analysis of the integrative review process lacked any difference in results for UR versus non-UR students at all admissions phases, even though the main difference in process was at the application stage. We see two possible reasons for this result as compared to the prior comparison of traditional versus algorithmic review processes; first, by the time integrative review was used, the top non-UR applicants may have achieved the level of preparedness of UR applicants, and second, the evolution of the review process may have removed more reviewer bias at the application phase. There is no way for us to distinguish between these possibilities (e.g., overprepared compared to pool and subsequent pool preparation catch-up vs. reviewer bias towards UR using earlier review processes). Regardless of the distinction between these two possibilities, we see an overall conclusion that the integrative review process demonstrated no admissions phase opportunity difference for UR versus non-UR applicants, yet a large percentage of UR applicants continue to be admitted to the program. We further point out that the possibility that our reviewers were biased towards UR applicants in earlier time periods suggests that an integrative process may be used to remove bias from other review processes, regardless of the direction of that bias.

The authors would like to note that studying sequential cohorts admitted using different review processes may be confounded by historical effects unrelated to changes in the review process. Examples include the COVID-19 pandemic, which moved admissions and recruitment processes to a virtual environment, biomedical science programs' gradual shift from GRE requirements in the application, and MCGSBS's removal of the application fee in 2020. Such systemic changes likely influenced the applicant pool, which may have affected these results.

### Structural changes that can support increased diversity during admissions

Our admissions process is a living framework that is open to change. Since the start of our program, we have recognized the importance of diversity in biomedical research [8]. Additionally, we understand that institutional and systematic barriers

may impact access to resources for applicants interested in our program. Our integrative approach provided us the opportunity to give credence to applicants' lived experiences without the impact of other potentially biased elements (e.g., GPA and CV) [2] and has shown successful recruitment of diverse trainees.

The standard review of applications emphasizes an applicant's past and utilizes their pedigree as a measure of their worth. While past is an important component, it can be a measure of many circumstances outside of the applicant's control, such as access to mentors and opportunities. The integrative review process reduces the emphasis on the applicant's past and places it instead on their present and future, shifting the focus from pedigree to potential. An applicant is allowed to delve into their past as much or as little as they wish. The story of who they were, are, and want to become is theirs to tell. This is a review system that is resistant to institutional and societal changes that may determine different priorities at different times. What we have found here is that an integrative review framework is able to maintain an equivalent admissions diversity index without the need to utilize any demographic information about the applicants.

We emphasize that the main goal of our admissions process is to recruit the best applicants in an equitable way. We do this by promoting applicant autonomy in assessment of their potential. An unexpected benefit is that this framework may also be more efficient than traditional or algorithmic application review. Efforts to increase admissions efficiency should first ensure equitable recruitment of applicants is present and that it is continued after admissions process changes. Streamlining admissions without affecting cohort diversity may be possible via many avenues. Other studies have introduced similar processes to mitigate reviewer biases and some have analyzed effects by UR status [45–47].

Furthermore, this work comes with the caution that no admissions metrics guarantee any type of future achievement. The application of each student is a cross-section of their current abilities, past successes, and possible potential at the time of applying. What is clear, however, is that certain metrics have excluded and continue to exclude excellent under-represented candidates. So, admissions must be approached with caution and clarity about how each application element will be considered and why it is used to evaluate a student. Even better would be to transparently detail admissions procedures and demographics of students throughout this process, as we present here, allowing students to make informed decisions as they approach the application cycle and programs and to identify strategies effective for meeting shared goals across institutions. This work aligns with the first recommendation of the DEIA CTSA Task Force Programmatic focus area: Develop Diversity, Equity, Inclusion, and Accessibility Conducive Training Environments, specifically by "providing infrastructural support to share methods and findings among learning communities" [10,48].

There are always other areas for continued improvement, including recruitment and retention of diverse trainees in biomedical science and increasing pipeline diversity by increasing access to resourced and targeted mentoring [11]. Other ways to specifically bolster ethno-racially diverse recruitment include partnering with historically black colleges/universities and minority-serving institutions [14]. Increasing diverse trainees in graduate school programs requires time, commitment, and multi-faceted approaches that do not stop at admittance [2].

## Limitations

CTS is one program within MCGSBS and thus has a shared application with other tracks, limiting control of application component collection. We were not able to be prospectively transparent to all applicants about our initial review of their PS alone. We did share this information with those that were provided an interview but agree that there should be transparency from the beginning [11]. There may be a lack of generalizability in our findings from this case study of our admissions process, as we are part of a centrally funded, highly specific PhD program. Still, there are aspects of our integrative framework that other biomedical science programs can incorporate from this case study. Using an integrative review may not have completely removed bias introduced in the application process; for example, blinding did not change the possible mention of mentor names and institutions in the PS, but it does contribute to a more equitable review of applications to help combat institutional and societal barriers.

Furthermore, our data do not include international applicants and may not be reflective of full cohort diversity. The TL1 grant, which is the major funding source for the students whose admissions data is included in this work, only allowed the

admission of domestic applicants until 2020, when our program acquired additional funding. MD-PhD students were also excluded as they are not admitted through our process but are reviewed by a separate admissions committee. Additionally, the NIH definition of "under-represented identity" has evolved during the past decade, and thus, earlier years' data may underestimate true past diversity present in graduate student cohorts.

We acknowledge limitations in the data analysis, namely the before-after type evaluation style and that individual cohorts did not experience multiple application review processes. Limitations, including the lack of a control group and temporal changes, limit the generalizability of these findings. Due to the nature of retrospectively analyzing a living process, we could not prospectively compare individual applicant performance across all review processes simultaneously. The limited sample size reflects the challenges of studying cohorts from specific PhD programs. Still, we have attempted to use pooling to mitigate this, as admissions processes are too individualized to study across institutions.

We realize that a process with seven reviewers per application is resource intensive, however we believe the admissions committee should represent the interests and opinions of all roles in the program. These include faculty, alumni, current students, and staff. We suggest at minimum at least one representative reviewer from each of these roles, however increasing this to even two opinions per role may allow for greater consensus. Individual admissions bodies should assess their resources and internal variance to best employ this strategy.

## Future directions

A follow-up study on the outcomes of cohorts admitted using the review framework introduced here is warranted. However, choosing appropriate outcome measures is controversial, as previously discussed. Follow-up studies of the integrative review cohort should include tangible and traditional success outcomes, but this will be program-specific. Another study of interest could assess what level of preparedness students felt necessary to demonstrate prior to applying, which could address the question of UR preparation versus reviewer preference for UR. Furthermore, qualitative assessments of why students accepted their admittance offer could highlight themes important to retention post-recruitment and areas for improvement other than admissions.

## Conclusion

One of the NIH/NCATS strategic planning initiative's goals for higher education is to increase diversity in graduate programs through recruitment, admission, and retention of a highly diverse cohort of students. This case study of one translational PhD program shows the evolution of its admissions process to support this goal. While early admission processes were successful, with high cohort diversity from the program's inception, today the admissions process furthers these efforts sustainably and unbiasedly through a review process that uses blinding and prioritizes the applicants' chosen narrative to emphasize their potential.

## Author contributions

**Conceptualization:** Minerva A Orellana, Carmen J. Silvano, Felicity T. Enders, Marina R. Walther-Antonio.

**Data curation:** Carmen J. Silvano, Ryan Wuertz, Jennifer L. Weisbrod.

**Formal analysis:** Danielle J. Beetler, Felicity T. Enders.

**Methodology:** Felicity T. Enders.

**Supervision:** Carmen J. Silvano, Ryan Wuertz, Lewis R. Roberts, Anthony J. Windebank.

**Visualization:** Marina R. Walther-Antonio.

**Writing – original draft:** Minerva A Orellana, Danielle J. Beetler.

**Writing – review & editing:** Minerva A Orellana, Danielle J. Beetler.

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
