## [Decision Letter · Decision Letter 0]

2 Aug 2024

Dear Dr. Orellana,

We look forward to receiving your revised manuscript.

Kind regards,

Jake Anders

Academic Editor

PLOS ONE

“This publication was supported by CTSA Grant Number TL1 TR002380 from the National Center for Advancing Translational Science (NCATS). Its contents are solely the reponsibility of the authors and do not necessarily represent the official views of the NIH.”

Reviewers' comments:

Reviewer's Responses to Questions

**Comments to the Author**

1. Is the manuscript technically sound, and do the data support the conclusions?

Reviewer #1: Partly

Reviewer #2: Partly

2. Has the statistical analysis been performed appropriately and rigorously?

Reviewer #1: Yes

Reviewer #2: No

3. Have the authors made all data underlying the findings in their manuscript fully available?

Reviewer #1: No

Reviewer #2: No

4. Is the manuscript presented in an intelligible fashion and written in standard English?

Reviewer #1: Yes

Reviewer #2: Yes

Reviewer #1: This paper describes the admissions process used by the MCGSBS CTS PhD program over the last two years, and presents results of the admissions process over the last 10 years.

In general, there is not enough attention paid to graduate admissions processes, and I laud the authors for taking a critical eye towards the admissions processes at their own school. However, based on the data presented, it is difficult to draw conclusions about what policies would be useful for other programs to implement.

General comments:

* The authors describe an interesting method of selecting applicants for interview, which was implemented over the last two admissions cycles: De-identifying the personal statements and selecting applicants for interview based only on this information, with the rest of the admissions packet only becoming available to interviewers at the time of the interview. However, the effects of this policy were not directly compared with the effects of other policies implemented in other years, and it is therefore very difficult to determine the effectiveness of the policy. To address this comment, I would want to see:

** A description of the admissions policies used in years prior to 2022-2023 and 2023-2024, how these policies differ from the one used in the last two cycles, and what the motivation is for changing the policies. Based on the data in this plot, the policies used from 2019-2021 appear to have similar effects as the policies used in 2022-2023. Why is the admissions committee no longer using the 2019-2021 policy? If the motivation is to seek effective alternatives to affirmative action because the law has now changed, state that explicitly and comparatively describe the admissions policy used in the last two years to the affirmative action admissions policies used in previous years. If there was no consistent admissions policy prior to the 2022-2023 admissions cycle, state that.

** Given that the percentage of underrepresented students in graduate programs has been increasing over the last ten years, I do not think it is appropriate to compare the effectiveness of the policy used in the last two admissions cycles to policies used more than 5 years ago.

** I suggest the authors' data on underrepresented applicants, interviewees, and admitted students would be easier to interpret if, instead of only presenting a plot of the raw data over the years (Figure 2), the authors reported the odds of an individual passing to the next stage of the process for each of the last ~5 years, comparing their admissions policy to the previous admissions policy. Look to the analysis in Barcelo et al (2021) for an example: https://link.springer.com/article/10.1007/s40596-020-01327-5

** An updated Figure 2 which uses the same labeling as in the text, or an update to the text to use the same language as in Figure 2. Currently, the text refers to cycles 2022-2023 and 2023-2024, while the labels in Figure 2 refer to single years. It is not clear to me which single year refers to when the new admissions policies were implemented, which makes the plot difficult to interpret.

* I am not sure what purpose the longitudinal data on admissions outcomes serves in the paper. The data presented show that the CTS PhD program has more effectively recruited underrepresented students than other programs around the country over the last 10 years. However, it's not clear to me what conclusions I am supposed to draw from this. The authors should either clarify what conclusions other admissions officers in other programs could reasonably draw from this data in order to improve their own admissions policies, or take out this longitudinal data.

Minor comments:

* The authors had a nuanced and detailed discussion of some of the motivations for diversifying graduate admissions in the first full paragraph on Page 3, which I greatly appreciated.

* I suggest reframing the second sentence of the discussion of CVs (last full paragraph on Page 4). Currently the authors sayy that the demonstration of prior research experience is "one of the most important qualities for a successful PhD program applicant." Instead, I suggest emphasizing that this is one of the qualities that admissions committees tend to place a large amount of weight on. This fits better with the tone set in the rest of the paragraph and I believe with the authors' goals in this paragraph (and paper).

* In the discussion of letters of recommendation (p. 4-5), the authors mention that demographic information about the letter writer can affect the way that they write the letter, which is true. However, it is also true that implicit bias can cause the same person to write a different letter for people of different demographics who performed at the same level, and even if the letters are identical, if they are about people in different demographic groups they are likely to be interpreted differently. For example, "X is an exceptional researcher!" can be interpreted as "X is an exceptional researcher [for their race]!" by a biased reader. The authors may wish to read and/or cite some of the following research on the topic:

Grimm et al 2020: Gender and racial bias in radiology residency letters of recommendation

Madera et al 2019: Raising doubt in letters of recommendation for academia: Gender differences and their impact

Biernat et al 2018: Effects of dyadic communication on race-based impressions and memory

Collins et al 2009: Stereotypes in the communication and translation of person impressions

* The discussion of what defines a "successful" graduate student is important. The authors definitions of success should be established early and used to guide the discussion of the compared policies and their outcomes. I would like to see some information about this concept in the introduction, along with a working definition of success in graduate school which the authors will use in their methods & results, with further discussion tying the concept of student success used in this paper to other concepts of student success used in the literature (& relevant complications) in the discussion section.

Reviewer #2: This is an interesting paper on an under-explored and important topic. I am keen to see evidence on this issue disseminated, which contributes strongly to my recommendation.

However, I currently need further persuasion that this paper provides sufficient supporting evidence for the conclusions that it reaches. It carries out a number of analyses that, while pertinent to understanding the diversity of the cohort of students on the case study program of study, are not relevant to the question of whether blinded review is helpful in achieving this aim, given that they do not extend to the years in which that blinded review has been introduced (analyses run from 2008-2018, blinded review is introduced in 2022). To be suitable for publication, it is vital that the paper is either re-focussed on assessing the evidence for the blinded review process — given that this is set up as its focus in the title and abstract — or re-framed such that the broader analyses are in support of the aims of the publication reflected in the title, abstract and introduction.

It is also fair to say that I do have concerns about the ability of the programme to provide sufficient data to provide supporting evidence of the type that is needed, simply because of its scale. Sample sizes in single years are very small, with the 100% under-represented applicants, interviewees and offer recipients in two years being an acute manifestation of this. Perhaps more could be done by pooling years or using moving averages (although this latter option is a challenge when looking at policy discontinuities), which the authors could consider. There may be other ways to address this concern, which I’m open to seeing, but this needs to be carefully considered.

In addition, at present the description of the statistical methods is limited and not consistently linked to the paper’s implied research question. I reproduce the entirety of the information on this in the methods section here to allow easy discussion of the parts that need more information.

First, what is not present in the methods section is what is implied by the abstract to be the primary analysis (and, indeed, the most directly relevant to understanding the changes in the process introduced in 2022-2023). The only detail about the construction of Figure 2 is in the notes to the figure itself and, as a result, there is no discussion of the method and its suitability to address the implied research question. The chart does show that in most years for which the analysis has been done the proportion of applicants who are under-represented is similar to the proportion of interviewees who are under-represented, and then generally the proportion of those offered a place who are under-represented is higher. However, this bounces around a lot because of the small sample sizes involved (as discussed above) and we can’t really learn much at all from 2021 or 2023 given that 100% of applicants (and, therefore, by definition also interviewees and offer recipients) are classified as under-represented. In the one cohort after the blinded review where we have the potential for variation in the proportion of interviewees and offer recipients who are under-represented, the rate of applicants and offer recipients who are under-represented are broadly consistent with earlier cohorts, while the rate among interviewees looks somewhat higher. This is consistent with a positive change due to the blinded review process — but is suggestive at best and seems unlikely to be a statistically significant difference given the numbers involved. Even ignoring the sample sizes involved, this is effectively a before-after type evaluation design and the limitations of evidence from such analyses should be discussed.

“Retrospective analyses of program trends from alumni cohorts compared to National CTS TL1 program cohort aggregates and NIH aggregate data for graduate students nationwide...”

More information on these data sources is needed at this point, including the measures that they include (you come back to this in a more ad-hoc way in the Results section, but it should be discussed in the Methods section). Clarity is also needed over the years for which this retrospective analysis is possible, i.e., why can’t it be any more recent? Especially as, without coming up to years in which the blinded review has been implemented, there should be no hint of an implication that this lends support to the hypothesis that the blinded review supports the recruitment and success of candidates from under-represented backgrounds. These successes — which I am not wishing to denigrate — appear mainly related to other features of the course.

“...was completed using Chi-Square analyses. A p value < 0.05 was considered significant.”

“Chi-Square analyses” and “analyses of program trends” are vague and I would encourage you at this point in the manuscript to spell out what exactly you are going on to do, i.e., compare demographic proportions in your program data to proportions of the same categories in the benchmark data that you have assembled. It’s also worth saying that with expected cell counts of below 5 in some of the categories (those on the program), there is a question as to whether a chi-squared test is appropriate rather than Fisher’s exact test.

“Secondary metrics were reported as mean time-to-degree from all alumni cohort years, while median publication counts were reported for each cohort year, as numbers of publications per student in each cohort year were not normally distributed.”

In a similar spirit to the previous point about the analysis of program data compared to benchmark data, there is a need to be explicit the question(s) that these analyses address and their relevance to the issue of the blinded review policy. Without this, there is a risk of this being misinterpreted as providing direct evidence of the effectiveness of the blinded review policy, whereas this the link is not clear to me at present.

**Do you want your identity to be public for this peer review?** For information about this choice, including consent withdrawal, please see our Privacy Policy

Reviewer #1: **Yes: ** Sonia Roberts

Reviewer #2: **Yes: ** Jake Anders

---

## [Author Response · Author response to Decision Letter 1]

28 Jan 2025

The authors thank Reviewers 1 and 2 for their thorough review and suggestions. As much of the structural content of the paper has changed, and the Methods and Results sections are completely rewritten, redline changes to reflect edits were not possible. Line numbers reflecting edits specific to each review comment if relevant have been included.

The manuscript has been adapted to PLOS ONE’s style requirements.

“This publication was supported by CTSA Grant Number TL1 TR002380 from the National Center for Advancing Translational Science (NCATS). Its contents are solely the responsibility of the authors and do not necessarily represent the official views of the NIH.”

We have included the Role of Funder statement within the cover letter.

There are legal restrictions within the data set and it cannot be publicly shared as is it is protected by the Family Educational Rights and Privacy Act (FERPA). There is a small sample size in some of the years so identities could be extrapolated from the information. This is stated in the Methods Data Analysis section beginning at line 186.

This has been completed.

Reviewer #1:

This paper describes the admissions process used by the MCGSBS CTS PhD program over the last two years, and presents results of the admissions process over the last 10 years.

In general, there is not enough attention paid to graduate admissions processes, and I laud the authors for taking a critical eye towards the admissions processes at their own school. However, based on the data presented, it is difficult to draw conclusions about what policies would be useful for other programs to implement.

General comments:

* The authors describe an interesting method of selecting applicants for interview, which was implemented over the last two admissions cycles: De-identifying the personal statements and selecting applicants for interview based only on this information, with the rest of the admissions packet only becoming available to interviewers at the time of the interview. However, the effects of this policy were not directly compared with the effects of other policies implemented in other years, and it is therefore very difficult to determine the effectiveness of the policy. To address this comment, I would want to see:

A description of the admissions policies used in years prior to 2022-2023 and 2023-2024, how these policies differ from the one used in the last two cycles, and what the motivation is for changing the policies. Based on the data in this plot, the policies used from 2019-2021 appear to have similar effects as the policies used in 2022-2023. Why is the admissions committee no longer using the 2019-2021 policy? If the motivation is to seek effective alternatives to affirmative action because the law has now changed, state that explicitly and comparatively describe the admissions policy used in the last two years to the affirmative action admissions policies used in previous years. If there was no consistent admissions policy prior to the 2022-2023 admissions cycle, state that.

Figure 1 has been modified to reflect admissions process and admissions team alongside a timeline of when each admissions process was used. A more detailed description of the new integrative review process has been added to the methods section (starting line 158). The admissions committee changed policies before affirmative action law changes as part of an evolving admissions betterment process. The committee has continually explored means to reduce bias in admissions. This analysis sought to demonstrate that the new admissions process continues to admit diverse cohorts while lessening reliance on application elements prone to bias.

Given that the percentage of underrepresented students in graduate programs has been increasing over the last ten years, I do not think it is appropriate to compare the effectiveness of the policy used in the last two admissions cycles to policies used more than 5 years ago.

The historical data presented in the article is included to provide 1) a comprehensive lens into cohort diversity across time since the program inception and 2) to allow higher sample size for better data testing. Although the data from students admitted using the traditional process is outdated, it provides a key point in showing that early review processes resulted in disparate odds of admissions for both under-represented and non-under-represented students in this program. The limitations of using this data have been addressed within lines 212-214. Furthermore, more recent data (through the most recently admitted cohort in 2024) has been included/added.

** I suggest the authors' data on underrepresented applicants, interviewees, and admitted students would be easier to interpret if, instead of only presenting a plot of the raw data over the years (Figure 2), the authors reported the odds of an individual passing to the next stage of the process for each of the last ~5 years, comparing their admissions policy to the previous admissions policy. Look to the analysis in Barcelo et al (2021) for an example: https://link.springer.com/article/10.1007/s40596-020-01327-5

Thank you for this important suggestion. We have reworked the analysis to reflect this within the constraints of available data. The results section, especially Tables 2 and 3 reflect this key change.

An updated Figure 2 which uses the same labeling as in the text, or an update to the text to use the same language as in Figure 2. Currently, the text refers to cycles 2022-2023 and 2023-2024, while the labels in Figure 2 refer to single years. It is not clear to me which single year refers to when the new admissions policies were implemented, which makes the plot difficult to interpret.

This has been addressed by the shift in analysis and the pooling of admissions data for years that used the same review process.

I am not sure what purpose the longitudinal data on admissions outcomes serves in the paper. The data presented show that the CTS PhD program has more effectively recruited underrepresented students than other programs around the country over the last 10 years. However, it's not clear to me what conclusions I am supposed to draw from this. The authors should either clarify what conclusions other admissions officers in other programs could reasonably draw from this data in order to improve their own admissions policies, or take out this longitudinal data.

Thank you for this critique. Table 1 has been modified to only present the data for this specific program as a means to assess diversity in enrollment over time. References to national data have been removed as they clouded the overarching narrative.

Minor comments:

The authors had a nuanced and detailed discussion of some of the motivations for diversifying graduate admissions in the first full paragraph on Page 3, which I greatly appreciated.

Thank you for the comment.

* I suggest reframing the second sentence of the discussion of CVs (last full paragraph on Page 4). Currently the authors say that the demonstration of prior research experience is "one of the most important qualities for a successful PhD program applicant." Instead, I suggest emphasizing that this is one of the qualities that admissions committees tend to place a large amount of weight on. This fits better with the tone set in the rest of the paragraph and I believe with the authors' goals in this paragraph (and paper).

Thank you for this suggestion, it has been adapted in lines 96-97.

* In the discussion of letters of recommendation (p. 4-5), the authors mention that demographic information about the letter writer can affect the way that they write the letter, which is true. However, it is also true that implicit bias can cause the same person to write a different letter for people of different demographics who performed at the same level, and even if the letters are identical, if they are about people in different demographic groups they are likely to be interpreted differently. For example, "X is an exceptional researcher!" can be interpreted as "X is an exceptional researcher [for their race]!" by a biased reader. The authors may wish to read and/or cite some of the following research on the topic:

Grimm et al 2020: Gender and racial bias in radiology residency letters of recommendation

Madera et al 2019: Raising doubt in letters of recommendation for academia: Gender differences and their impact

Biernat et al 2018: Effects of dyadic communication on race-based impressions and memory

Collins et al 2009: Stereotypes in the communication and translation of person impressions

Thank you for the insightful comment and the topical literature suggestions. These have been incorporated and the language has been changed in lines 107-109.

* The discussion of what defines a "successful" graduate student is important. The authors definitions of success should be established early and used to guide the discussion of the compared policies and their outcomes. I would like to see some information about this concept in the introduction, along with a working definition of success in graduate school which the authors will use in their methods & results, with further discussion tying the concept of student success used in this paper to other concepts of student success used in the literature (& relevant complications) in the discussion section.

Thank you for the structural suggestion. The description of graduate student ‘success’ has been moved into the introduction (lines 132-144) and the narrative of the paper ahs been modified to remove focus on these outcome-based metrics in this paper. Additions to the future directions section elaborate further on what kind of study could be done with future data from the cohorts of students admitted using this new policy (lines 311-316). 

Reviewer #2:

This is an interesting paper on an under-explored and important topic. I am keen to see evidence on this issue disseminated, which contributes strongly to my recommendation.

However, I currently need further persuasion that this paper provides sufficient supporting evidence for the conclusions that it reaches. It carries out a number of analyses that, while pertinent to understanding the diversity of the cohort of students on the case study program of study, are not relevant to the question of whether blinded review is helpful in achieving this aim, given that they do not extend to the years in which that blinded review has been introduced (analyses run from 2008-2018, blinded review is introduced in 2022). To be suitable for publication, it is vital that the paper is either re-focussed on assessing the evidence for the blinded review process — given that this is set up as its focus in the title and abstract — or re-framed such that the broader analyses are in support of the aims of the publication reflected in the title, abstract and introduction.

Thank you for this important critique. The paper has been substantially reworked to provide analyses that better assess the new review framework against previous processes.

It is also fair to say that I do have concerns about the ability of the programme to provide sufficient data to provide supporting evidence of the type that is needed, simply because of its scale. Sample sizes in single years are very small, with the 100% under-represented applicants, interviewees and offer recipients in two years being an acute manifestation of this. Perhaps more could be done by pooling years or using moving averages (although this latter option is a challenge when looking at policy discontinuities), which the authors could consider. There may be other ways to address this concern, which I’m open to seeing, but this needs to be carefully considered.

To address sample size concerns, data has been pooled across years using the same review framework. This substantially improved the quality of evidence and ability to assess the new review framework.

In addition, at present the description of the statistical methods is limited and not consistently linked to the paper’s implied research question. I reproduce the entirety of the information on this in the methods section here to allow easy discussion of the parts that need more information.

First, what is not present in the methods section is what is implied by the abstract to be the primary analysis (and, indeed, the most directly relevant to understanding the changes in the process introduced in 2022-2023). The only detail about the construction of Figure 2 is in the notes to the figure itself and, as a result, there is no discussion of the method and its suitability to address the implied research question. The chart does show that in most years for which the analysis has been done the proportion of applicants who are under-represented is similar to the proportion of interviewees who are under-represented, and then generally the proportion of those offered a place who are under-represented is higher.

However, this bounces around a lot because of the small sample sizes involved (as discussed above) and we can’t really learn much at all from 2021 or 2023 given that 100% of applicants (and, therefore, by definition also interviewees and offer recipients) are classified as under-represented. In the one cohort after the blinded review where we have the potential for variation in the proportion of interviewees and offer recipients who are under-represented, the rate of applicants and offer recipients who are under-represented are broadly consistent with e

---

## [Decision Letter · Decision Letter 1]

25 Feb 2025

Dear Dr. Orellana,

Thank you for submitting your manuscript to PLOS ONE. After careful consideration, we feel that it has merit but does not fully meet PLOS ONE’s publication criteria as it currently stands. Therefore, we invite you to submit a revised version of the manuscript that addresses the points raised during the review process.

ensuring the discussion a) discusses interpretation of the statistical modelling, b) draws out implications of the findings more broadly for readers;whether you are happy with the suggestion to contextualise the paper as a case study, as recommended;reflecting on the full and short title and ensuring they are both well-aligned with the contribution of the paper.

If addressed appropriately, I would not anticipate this paper requiring another round of revisions.

We look forward to receiving your revised manuscript.

Kind regards,

Jake Anders

Academic Editor

PLOS ONE

Journal Requirements:

Reviewers' comments:

Reviewer's Responses to Questions

**Comments to the Author**

Reviewer #1: (No Response)

Reviewer #2: (No Response)

2. Is the manuscript technically sound, and do the data support the conclusions?

Reviewer #1: Partly

Reviewer #2: Yes

3. Has the statistical analysis been performed appropriately and rigorously?

Reviewer #1: Yes

Reviewer #2: Yes

4. Have the authors made all data underlying the findings in their manuscript fully available?

Reviewer #1: Yes

Reviewer #2: No

5. Is the manuscript presented in an intelligible fashion and written in standard English?

Reviewer #1: Yes

Reviewer #2: Yes

Reviewer #1: The authors have significantly improved the paper, and I commend them for their efforts to address all of the previous comments. I only have two comments that I think are truly vital to be addressed before the paper is ready for publication.

1. The authors analyzed the odds of a student reaching a particular phase in the admissions process given certain characteristics (e.g. URM status), which makes the data much easier to interpret than in the previous version of the paper. However, the discussion of these results is very brief. The paper (and its readers!) would benefit substantially from a subsection in the Discussion talking about the implications of these results and spelling out the conclusions for the readers.

2. I still find myself somewhat unsure what the reader is meant to learn from this data so that they can improve their university's admissions processes. However, I have a much better guess from this version of the paper due to the authors' significant improvements. Is the take-home message that the de-identified PS evaluation produces the same results as an algorithmic admissions procedure, even though it takes significantly less time to perform? If that is the case, the authors should state that explicitly and present their case study as evidence for using de-identified personal statements to decrease the amount of time required for the first round of application reviews. Otherwise, I am not sure what conclusions I am meant to draw as a reader.

There is one additional thing I would like to see addressed that is less crucial but would help support the overall conclusions of the paper, assuming that I am now interpreting the paper correctly. What was the variance on the 7 application reviewers' scores of the de-identified PSs? I am concerned that most graduate programs would not be able to support having 7 reviewers for each de-identified PS. Is it the case that 7 reviewers were actually needed to come to a consensus, or could a consensus reasonably have been reached with (say) 3 reviewers? Knowing the answer to this question could help the authors discuss how widely applicable their case study may be.

Reviewer #2: I am grateful to the reviewers for a thorough revision of the manuscript, taking on board the range of points raised and critiques of the limitations of evidence that can be drawn from the case study that this effectively presents. I have a few remaining points that I think would be helpful to address in order to maximise the value of this paper to readers, as well as a few minor points that I spotted while reading.

The short title of the paper and its full title are quite strikingly different. Both have their merits so I think it’s worth reflecting on what contribution the paper is making that you want to foreground in the title. For me, showcasing an example of an integrative application review process is definitely part of it. I would suggest something like “Integrative application review in graduate school PhD admissions: a case study of efficiently minimizing bias while maximizing the student narrative” as the full title, with a short title that drops the bits after the colon, but that’s just one fairly quick go at that.

Drawing somewhat on an embedded point of that last one, I think you could be more explicit that this is, effectively, a case study. Case studies have strengths and limitations and thinking of the study in this way will help people interpret the results appropriately, noting that this will imply attention to context in seeking lessons for other contexts.

Relatedly, the discussion section could helpfully do more to foreground the implications of the paper’s finding for others running higher education admissions exercises (especially graduate biomedical programmes, although perhaps not exclusively). In service of drawing out such implications, you could link your findings to other areas of the literature on higher education admissions that have sought to understand how aspects of the admissions process affect potential biases. (Full disclosure: I wrote the paper I’m about to mention and you should not feel under any obligation to cite it but, perhaps inevitably, connections to one’s own work will come to mind!) Anders (2014) found that introducing an aptitude test in the admissions process for a highly competitive undergraduate course reduced the number of applicants who needed to be interviewed without affecting gender or socioeconomic representation in those selected, which mirrors your own finding of being able to streamline admissions using your proposed framework without negatively affecting diversity. There are other papers that have relevant findings against which yours could be compared, likely with closer context.

The paper’s conclusions has become a little disjoint from the rest, where the focus has become more on the overall nature of the ‘integrative’ application review process, while this just mentions blinded review in isolation. I think having embedded it within a wider framework has been helpful in explaining what you have done and its potential lessons, so would encourage you to ensure the conclusion section aligns with this (potentially also checking for similar issues elsewhere).

Minor point: Claims in the introduction/review sections are mostly well-evidenced but the sentence “We agree with prior cautions that this style is most prone to biases that may limit desired demographic changes and equitable access for all applicants.” on lines 117-119 would benefit from a citation to help people trace evidence of the type discussed.

References

Anders, Jake, (2014), Does an aptitude test affect socioeconomic and gender gaps in attendance at an elite university?, No 14-07, DoQSS Working Papers, Quantitative Social Science - UCL Social Research Institute, University College London, https://EconPapers.repec.org/RePEc:qss:dqsswp:1407.

**Do you want your identity to be public for this peer review?** For information about this choice, including consent withdrawal, please see our Privacy Policy

Reviewer #1: **Yes: ** Sonia Roberts

Reviewer #2: **Yes: ** Jake Anders

---

## [Author Response · Author response to Decision Letter 2]

3 Apr 2025

The authors thank the Reviewers again for their continued comments that have greatly benefitted this manuscript. The suggestions have been integrated into this second revision. If relevant, line numbers reflecting edits specific to each reviewer comment have been included.

Reviewer #1:

The authors have significantly improved the paper, and I commend them for their efforts to address all of the previous comments. I only have two comments that I think are truly vital to be addressed before the paper is ready for publication.

1. The authors analyzed the odds of a student reaching a particular phase in the admissions process given certain characteristics (e.g. URM status), which makes the data much easier to interpret than in the previous version of the paper. However, the discussion of these results is very brief. The paper (and its readers!) would benefit substantially from a subsection in the Discussion talking about the implications of these results and spelling out the conclusions for the readers.

Thank you for this critique. The results sections describing the data (Lines 222-228 and 230-239) and the discussion (Lines 263-282) have been edited to expand our interpretation of these results and the implications have been further detailed later in the discussion (Lines 297-313).

2. I still find myself somewhat unsure what the reader is meant to learn from this data so that they can improve their university's admissions processes. However, I have a much better guess from this version of the paper due to the authors' significant improvements. Is the take-home message that the de-identified PS evaluation produces the same results as an algorithmic admissions procedure, even though it takes significantly less time to perform? If that is the case, the authors should state that explicitly and present their case study as evidence for using de-identified personal statements to decrease the amount of time required for the first round of application reviews. Otherwise, I am not sure what conclusions I am meant to draw as a reader.

The authors appreciate the perspective shared by the reviewer. To more directly state the purpose of this process and its key takeaways, we have added a section to the discussion (Lines 307-325) as well as edited the conclusion to provide a more cohesive and consistent message (Lines 373-378). The authors hope that these (and other changes made in this revision) have helped to hone in on the central point that this type of process promotes the prioritization of the student’s own narrative while minimizing potential bias.

There is one additional thing I would like to see addressed that is less crucial but would help support the overall conclusions of the paper, assuming that I am now interpreting the paper correctly. What was the variance on the 7 application reviewers' scores of the de-identified PSs? I am concerned that most graduate programs would not be able to support having 7 reviewers for each de-identified PS. Is it the case that 7 reviewers were actually needed to come to a consensus, or could a consensus reasonably have been reached with (say) 3 reviewers? Knowing the answer to this question could help the authors discuss how widely applicable their case study may be.

The authors appreciate the opportunity to address this point to clarify the conclusions of the paper, Variance cannot be determined as rubrics were used to gather qualitative feedback – this has been clarified in the methods section (Lines 158-178). The question of how many reviewers are needed to reach consensus is a good one, our process includes multiple program stakeholders to participate in the admissions committee. Although we cannot recommend a specific number, we have included more specific details about the application review process, including the makeup of the 7 reviewers for clarity (Lines 163-164). In the limitations, we have expanded on the issue of resource intensity/variance to suggest approaches that may be more widely applicable (Lines 357-362).

Reviewer #2:

I am grateful to the reviewers for a thorough revision of the manuscript, taking on board the range of points raised and critiques of the limitations of evidence that can be drawn from the case study that this effectively presents. I have a few remaining points that I think would be helpful to address in order to maximise the value of this paper to readers, as well as a few minor points that I spotted while reading.

The short title of the paper and its full title are quite strikingly different. Both have their merits so I think it’s worth reflecting on what contribution the paper is making that you want to foreground in the title. For me, showcasing an example of an integrative application review process is definitely part of it. I would suggest something like “Integrative application review in graduate school PhD admissions: a case study of efficiently minimizing bias while maximizing the student narrative” as the full title, with a short title that drops the bits after the colon, but that’s just one fairly quick go at that.

Thank you for this insightful comment. The short and full titles have been edited accordingly. (Lines 1-3 and Headers)

Drawing somewhat on an embedded point of that last one, I think you could be more explicit that this is, effectively, a case study. Case studies have strengths and limitations and thinking of the study in this way will help people interpret the results appropriately, noting that this will imply attention to context in seeking lessons for other contexts.

Per the suggestion to more appropriately label this work a case study, references to clarify the study type have been made throughout (Lines 1, 25, 150, 253, 337, 339, 374-375). A discussion of the limitations of this work being a case study has been added to the limitations (Lines 336-342)

Relatedly, the discussion section could helpfully do more to foreground the implications of the paper’s finding for others running higher education admissions exercises (especially graduate biomedical programmes, although perhaps not exclusively). In service of drawing out such implications, you could link your findings to other areas of the literature on higher education admissions that have sought to understand how aspects of the admissions process affect potential biases. (Full disclosure: I wrote the paper I’m about to mention and you should not feel under any obligation to cite it but, perhaps inevitably, connections to one’s own work will come to mind!) Anders (2014) found that introducing an aptitude test in the admissions process for a highly competitive undergraduate course reduced the number of applicants who needed to be interviewed without affecting gender or socioeconomic representation in those selected, which mirrors your own finding of being able to streamline admissions using your proposed framework without negatively affecting diversity. There are other papers that have relevant findings against which yours could be compared, likely with closer context.

The authors appreciate the suggestion. A paragraph discussing implications for similarly minded program administrations has been added to the discussion and includes references to similar efforts/results (Lines 307-313).

The paper’s conclusions has become a little disjoint from the rest, where the focus has become more on the overall nature of the ‘integrative’ application review process, while this just mentions blinded review in isolation. I think having embedded it within a wider framework has been helpful in explaining what you have done and its potential lessons, so would encourage you to ensure the conclusion section aligns with this (potentially also checking for similar issues elsewhere).

Thank you for this comment. The authors have edited the manuscript throughout to sharpen the focus to ‘integrative review using blinding’ rather than interchanging the terms blinded review and integrative review. As previously, the conclusion has been revised to provide better overall clarity and cohesion (Lines 373-378).

Minor point: Claims in the introduction/review sections are mostly well-evidenced but the sentence “We agree with prior cautions that this style is most prone to biases that may limit desired demographic changes and equitable access for all applicants.” on lines 117-119 would benefit from a citation to help people trace evidence of the type discussed.

Thank you for this catch. A citation to support this evidence has been added (Line 120).

---

## [Editor Report · Decision Letter 2]

8 Apr 2025

Integrative review in PhD admissions: a case study of efficiently minimizing bias while maximizing the student narrative

PONE-D-24-22191R2

Dear Dr. Orellana,

We’re pleased to inform you that your manuscript has been judged scientifically suitable for publication and will be formally accepted for publication once it meets all outstanding technical requirements.

Kind regards,

Jake Anders

Academic Editor

PLOS ONE
---

## [Editor Report · Acceptance letter]

PONE-D-24-22191R2

PLOS ONE

Dear Dr. Orellana,

I'm pleased to inform you that your manuscript has been deemed suitable for publication in PLOS ONE. Congratulations! Your manuscript is now being handed over to our production team.

Kind regards,

on behalf of

Prof. Jake Anders

Academic Editor

PLOS ONE